# A Spatio-Temporal Autowave Model of Shanghai Territory Development

**Natalia Levashova [1,]*, Alla Sidorova [2], Anna Semina [2] and Mingkang Ni [3]**

[1]    Department of Mathematics, Faculty of Physics, Lomonosov Moscow State University,
       Moscow 119991, Russia
[2]    Department of Biophysics, Faculty of Physics, Lomonosov Moscow State University, Moscow 119991, Russia
[3]    Department of Mathematics, East China Normal University, Shanghai, China
[*]    Correspondence: natasha@wanaku.net; Tel.: +7-916-134-1148

**Abstract:** A spatio-temporal model of megacity development that treats the megacity as an active medium is presented. From our point of view, it is advisable to consider the process of urban ecosystem development from the standpoint of the theory of autowave self-organization in active media. According to this concept, the urban ecosystem is considered as interacting with each other's natural and anthropogenic subsystems with significant heterogeneity of areas affected by human intervention and urban geobiocoenoses. The model is based on the general principles of active medium dynamics; therefore, it is universal for any object to be considered an active medium. The only difference when using the model to predict the development of urban ecosystems in countries with different socio-economic and political prerequisites is the variety of parameters included in the model, i.e., the activation parameter, the autowave process inhibitors, and the characteristic scales of the activator and inhibitor. The model was tested on the example of Moscow expansion in the period of 1952–1968 and showed good agreement with the map data. By means of the model, a prediction of Shanghai and surrounding territory development until 2030 was made.

**Keywords:** active media; activator; inhibitor; autowave self-organization; urban ecosystem

## 1. Introduction

In the context of the historical dynamics of *Homo sapiens*, structural spatial heterogeneity gradually developed in the anthroposphere due to the formation of cities and related adjacent territories (urban ecosystems (UES)). These systems are characterized by high-speed population growth and the concentration of residential, industrial, commercial, and other facilities, as well as communications. The more complex the structure of the system, the greater the number of possible stable states in it. This property is called multistability. The study of UES development features allows for predicting the evolution of territories ensuring effective interaction and balanced development of all spheres of life. With cities being complex nonlinear systems, mathematical modelling proposing acceptable paths of their development is difficult. For example, models with a Gaussian-type population density distribution, which are widely used to describe population density in cities [1,2], have a significant drawback in the additivity of the model parameters, while urban systems are non-uniform, non-linear systems that are characterized by the non-additivity principle. The authors of [3] proposed a model of a polycentric system, and a synergistic approach describing how UES development is embodied in the Lotka–Volterra models [4]. The application of fractal theory to describe the morphology of urban patterns [5] uses analogies between the urban pattern structure and fractal structures, thereby taking into account the spatial inhomogeneity. Much better results are shown by modern models based on the theory of cellular automata [6–11]. By means of these models, it became possible to simulate the growth

processes of different cities all over the world and to offer forecasts of their development [12–14]. Also, economics-based models like the hedonic pricing model are to be mentioned [15].

Here, we offer a different approach based on the theory of active media. This theory is most widely used to describe biological systems, such as a models of excitation processes in nerve cells, [16] dynamics of blood coagulation, and pulse shape observed in the myocardium [17]. From our point of view, the process of UES development can be considered as autowave self-organization in active media. Processes in UES can be compared with Belousov–Zhabotinsky reactions [18,19] occurring in a thin layer of liquid. In the absence of mixing, one can observe colored concentric rings (autowaves) propagating from active centers (pacemakers) along the liquid surface. These pacemakers are spontaneous centers of oscillations arising due to the inhomogeneity of the medium. The system is a two-dimensional distributed active medium in which the point oscillators interact in succession with each other. As other examples of stationary autowaves, we can cite models of color spot formation on the wings of butterflies and animal skins [20].

According to our concept, urban ecosystems (UES) are considered to be interacting with each other natural and anthropogenic subsystems with significant heterogeneity of areas affected by human intervention and urban geobiocoenoses. The UES pacemakers are the distribution and density of the population, as well as built-up areas. As a result of an increase in population, the density of buildings increases, as do industrial enterprises, the length of communications, trade structures, and traffic. Moreover, growing cities tend to merge into common systems, resulting in a change in geobiocoenosis areas and in the quantitative and qualitative composition of trophic networks. This, in turn, promotes population growth. Thus, the most active pacemaker captures the entire space of the UES, and the process becomes autocatalytic.

One of the most common and, at the same time, fairly simple models describing biological systems is the FitzHugh–Nagumo system [21–23]. This is a system of two parabolic equations of activator–inhibitor type for autowave propagation in a homogeneous environment. To create a model describing such a complex and nonlinear object as UES, we made modifications to this system, changing it so that it could describe not only propagating fronts but also stationary solutions with large gradients at the boundaries of barriers (natural geobiocoenoses that prevent autowave motion). In addition, a cross-product component was added to the equation for the activator, enhancing the feedback between the activator and the inhibitor.

## 2. Materials and Methods

In order to effectively apply the activator–inhibitor model for an adequate description of urban ecosystems, it is necessary to investigate thoroughly the reasons for their growth and the processes that inhibit this growth. While the density of the population and buildings is an obvious activator of urban ecosystem development, the inhibitor is determined by the policies determining urban planning and is different for each country [24,25]. Thus, any study should be preceded by the choice of an object and the determination of its individual characteristics.

### 2.1. Object of Study

Shanghai is one of the most dynamically developing regions of China, so it was chosen as the object of modeling. It is a shining representative of modern megalopolises, the rapid development of which is associated with an increase in the number of industrial enterprises. Back in the 1930s, Shanghai bypassed Guangzhou in terms of population; it still remains the largest city in China, and for many decades it has been among the world's top 10 agglomerations. This is the result of the political and economic reforms that have occurred in China since 1978 [26]. Currently, Big Shanghai is a typical monocentric agglomeration including 15 urban and 5 suburban areas. Over the past 50 years, its administrative borders have changed many times, which has led to an increase in the Shanghai territory by almost 10 times. During the same period, the population of Shanghai increased by 10.2 million people [27], which caused increased demand for housing.

In 1995, the National people's congress standing committee announced a new housing program, the "national project for comfortable housing" (hereinafter, "the Project") for major cities of China, including Beijing and Shanghai. One of the main tasks of the Project was to sell public housing to persons with middle and low incomes, as well as to involve the population's savings in housing investments. The main component of the reform was a system of Housing Subsidy Fund (HSF) creation. By the end of 1996, 98% of Shanghai's workers and employees were covered by the HSF system [28]. These reforms caused a wave of urbanization but at the same time had a number of negative consequences: The active growth in the number and density of the population and the expansion of urbanized areas due to natural geobiocoenosis. Generally, the expansion of urban areas in China in 2000–2005 amounted to 40.42%, which is much more than that in 2005–2010 (14.86%) and is related to the demographic policy of the Chinese government [29].

During the period from 1996 to 2000, the increase in housing was due to the allocation of 50 million square meters of land adjacent to the urban area for residential development [30]. Territorial expansion occurred in two stages: in 2000–2005, 160.61 km$^2$ was added to the South-West and in 2005–2010, 78.56 km$^2$ was added to the South-East. As a result of the acceleration of Shanghai's urbanization process, in 2000–2010, the population increased by 37.5% and its territory increased by 239.19 km$^2$. The population size and density has been constantly increasing in Shanghai (Chart 1), which has steadily entailed the expansion of its boundaries.

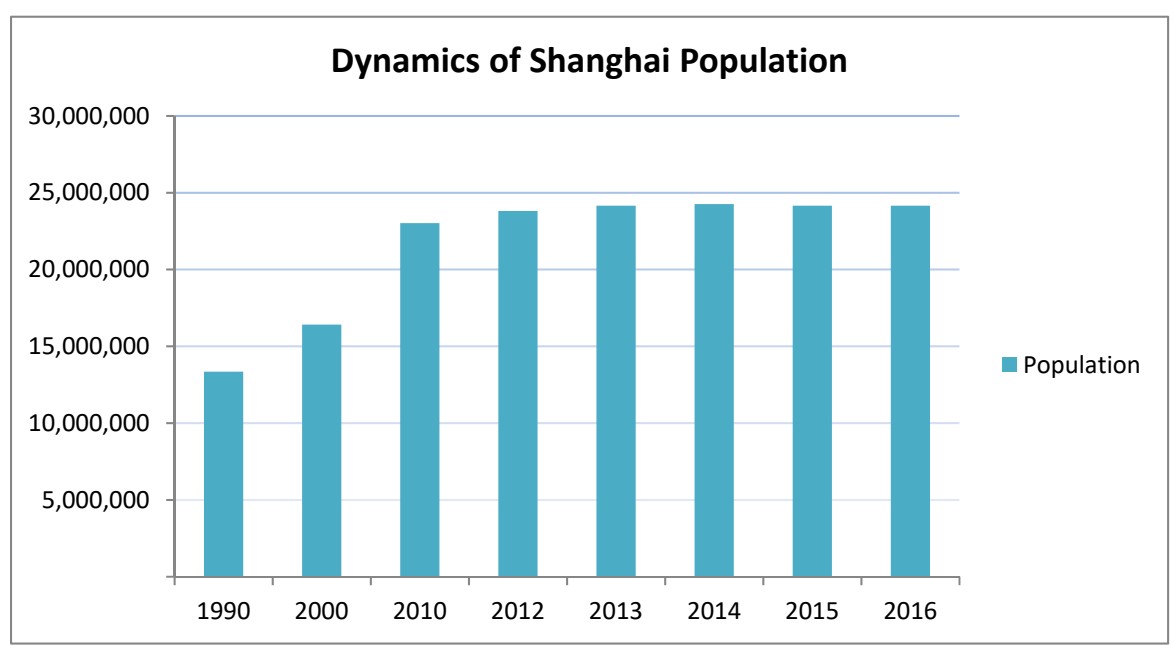

**Chart 1.** Population of Shanghai [27].

In the period of 2000–2010, the development of Shanghai was most active, which was associated with the presence of the social program for housing [31]. Shanghai expansion during this period was due to an increase in the density of development. Another important aspect is that during this period there was a trend of significant increase in the development rate of Shanghai Central in comparison with remote areas: An increase of 7 times for the period of 2000–2005 and of 5.5 times for the period of 2005–2010 (Charts 2 and 3). This trend was associated with the availability of social benefits for housing, which allowed buying realty in expensive areas of Shanghai. The average population density was 2059 people per km$^2$, and in the central part it was 3854 people per km$^2$ [27].

The directions of Shanghai expansion are South-East, South-West, and South due to the geographical location of the metropolis and to the industrial and economic growth of its central administrative region, Pudong [32].

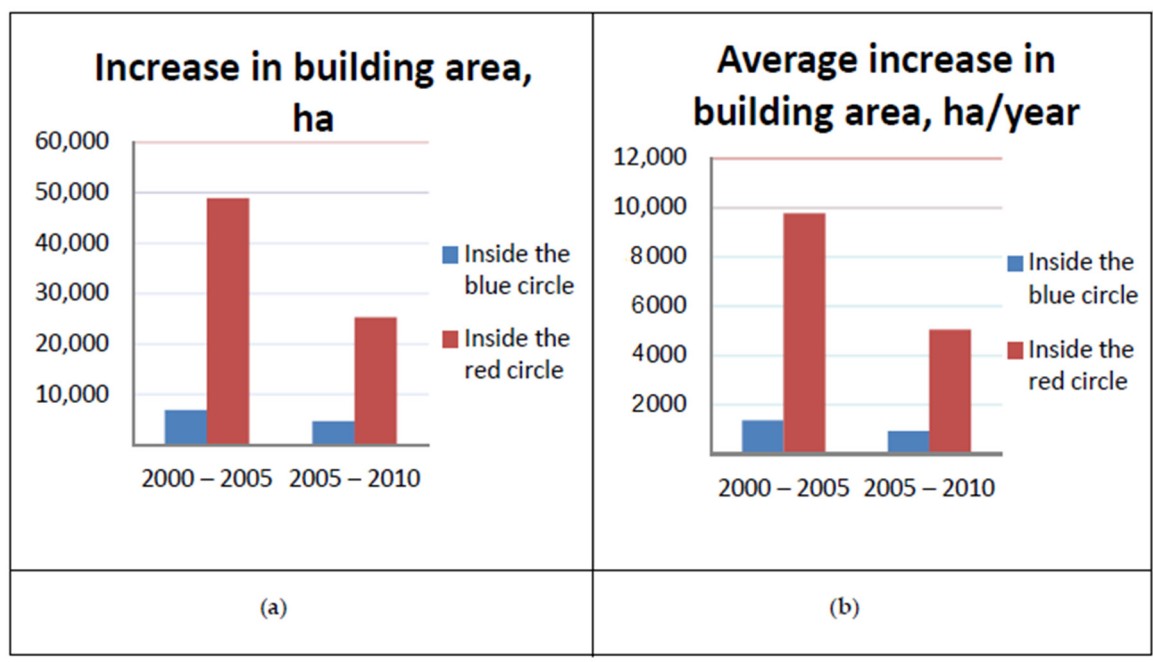

**Chart 2.** The dynamics of Shanghai housing development in 2000–2010 [31]: (**a**) The nominal increase in building area (ha) and (**b**) the average rate of development (ha/year).

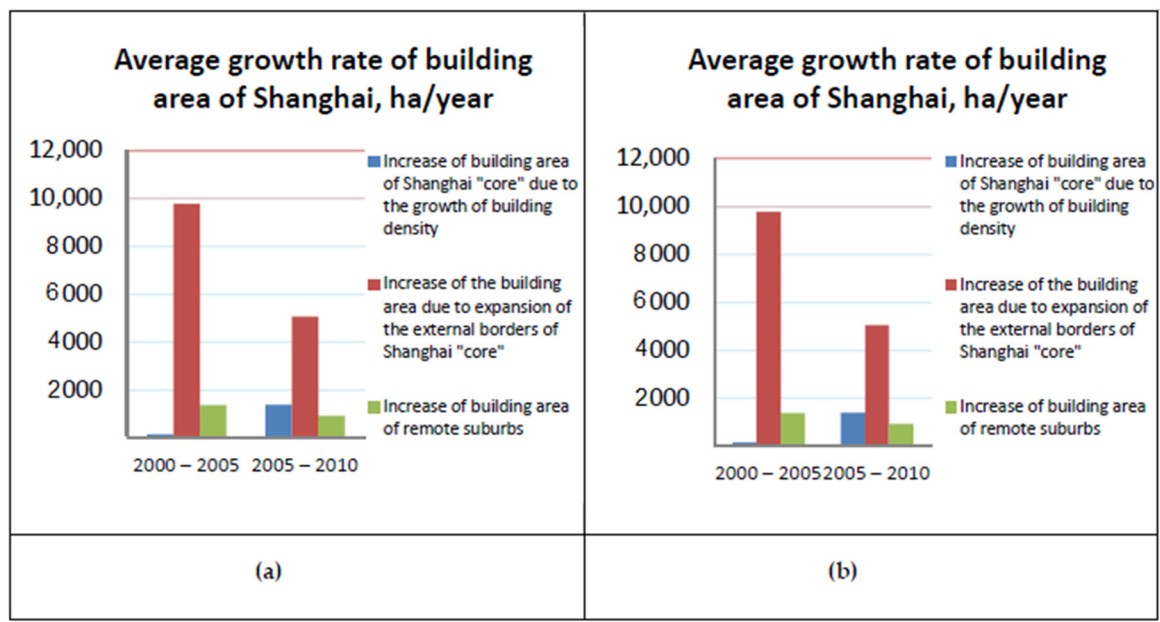

**Chart 3.** Dynamics of Shanghai development in 2000–2010 [32].

An analysis of socio-economic factors showed that in parallel with the population growth, there was growth in Shanghai's territory in the period 2000–2010 [32,33]. The demand for life quality increased, causing even more intensive use of land to meet these needs [34,35]. A study of the mechanisms affecting the territorial expansion of Chinese cities was conducted in [32] using GIS and GPS technology and high-resolution aerial photographs. According to the data obtained, socio-demographic factors (population size and density) were recognized as the main factors affecting the expansion of Chinese cities (and Shanghai in particular) and should therefore be taken into account to optimize the residential, industrial, and office building distribution. However, these factors also have an impact on natural ecosystems, as it is known that the expansion of cities results in changes in the

structure and functionality of adjacent territories, geobiocoenoses, and urban biocoenoses (landscape, biogeochemical and hydrological cycles, trophic networks, population dynamics) [36–40].

Therefore, at present, the Shanghai government is taking measures to limit the population total number and local density by 2030 [27]. The strategic development plan for Shanghai should include the following main tasks: The coordinated development of Shanghai and its suburbs, the preservation of natural geobiocoenoses, and the improvement of urban infrastructure.

The aim of our study is to build an autowave model for Shanghai development taking into account the basic mechanisms of this process: The spatial and socio-economic feasibility of development, the dynamics of housing cost, the proportion of land under agriculture, and the geographical features of adjacent territories.

Historically, cities in China developed as expanding circles from a common "core". In the period from 2000 to 2010, the expansion of Shanghai's borders was mainly due to the addition of agricultural and coastal land [32]. Now, the population density in the remote suburbs is 600–1000 people/km$^2$. It is obvious that development area can be chosen as the activator of the model.

The Choice of the Inhibitor

Studying atmospheric factors like precipitation and temperature in Shanghai and the surrounding areas [41,42] showed that, in general, they did not have an impact on Shanghai border expansion. Therefore, the most significant factors defining territory expansion are the number and density of potential buyers of housing and its cost. In 2015, China's official Xinhua news Agency reported that "by 2035, the permanent population in Shanghai will be controlled at about 25 million people, and the total amount of land available for construction will not exceed 3200 square kilometers" (to combat the so-called "big city disease"). This caused a rise in housing cost. The increase in housing cost, in turn, leads to lower paying capacity and, consequently, to lower profits for building companies, forcing them to limit development rates. Therefore, housing cost was chosen as the inhibitor of the model.

*2.2. Theoretical Background of a Spatio-Temporal Model of Shanghai Development*

To describe an urban ecosystem as active medium, the authors proposed a system of equations based on the modified Fitzhugh–Nagumo system [21–23,43–46]:

$$\frac{\partial u}{\partial t} - D_u \left( \frac{\partial^2 u}{\partial x^2} + \frac{\partial^2 u}{\partial y^2} \right) = -\frac{1}{T^*} u(u - \alpha(x,y))(u - 1) - \frac{1}{T^*} uv,$$
$$\frac{\partial v}{\partial t} - D_v \left( \frac{\partial^2 v}{\partial x^2} + \frac{\partial^2 v}{\partial y^2} \right) = \frac{1}{T^*}(-v + \gamma(x,y)u). \tag{1}$$

Here, $u$ is a dimensionless value equal to the share of built-up area per square kilometer (system activator); $v$ is the 1 m$^2$ residential area cost (share of maximum possible), a dimensionless value (system inhibitor); $T^*$ is a characteristic time scale (1 year); $\gamma(x,y)$ is a dimensionless parameter showing the dependence of the m$^2$ housing cost on the development district; $D_u$ is the rate of building area growth, ha/year; and $D_v$ is the rate of high housing cost spreading to low urbanized areas, ha/year.

The activator in this model possesses the property of bistability; that is, it can take one of two stable states. The inhibitor of the model is an excitable element that moves out of equilibrium as a result of a change in activator steady states. In the model, we define two steady states for the activator of urban development: High and low urbanization. The first is characterized by dense buildings and dense communications, while the second by the lack of urban infrastructure. As is known, a nonlinearity of cubic type, like in Equation (1), is best suited for describing bistable elements [47]. The stable states of the system are determined by the real roots of the cubic polynomial at the right-hand side of the equation. In the case where there are three real roots, the largest and the smallest correspond to steady states. The presence of three real roots is determined by the governing parameter $\alpha(x,y)$ [43–46]; that is, the property of the medium. A medium supporting two stable states of a bistable element is called excitable. The system solution tends to one of the stable roots when t→+∞ and to the other when

t→−∞. Thus, a region with a large gradient (autowave front) is formed between these roots, capable of propagating in the excitable medium. If the cubic equation has a single real root, then the front cannot form. In this case, the system acquires the non-excitable property and is characterized by one stable state. The inclusion of non-excitable areas (barriers) in the area of system consideration leads to the front stopping at the boundaries of barriers. As Shanghai is a flood plain without obvious changes in altitude and gradient, the main landscape features that we must take into account as barriers are large water reservoirs.

The existence theorem for moving front solutions of FitzHugh–Nagumo-type systems of equations was proved in [48]. The existence conditions for stable solutions with a large gradient at the boundaries of media discontinuities were formulated in [49,50].

### 2.3. The Study Methodology

In order to describe a real object, we have to use cartographic data when choosing a computational domain and use data on the spatial distribution of the activator and inhibitor at the initial moment of time. We must also set the values and distribution of the parameters $\alpha(x,y)$, $Du$, $Dv$, and $\gamma(x,y)$ of Equation (1) that, on the one hand, correspond to real data, and on the other hand, ensure the fulfillment of the existence conditions for an autowave solution [48–50].

#### 2.3.1. Map Data Processing

To analyze the cartographic data, the authors created a C-Sharp code, which creates a raster data model based on the maps. For this purpose, points are plotted on the map that differ in color depending on the properties of the territory (for our needs, this was either the density of the building, or the rate of growth in housing cost, or the presence of water barriers). Then, each color is associated with a certain numerical value of the parameter included in the model. These values are stored in binary arrays and then used for numerical solution of Equation (1).

The numerical solution was determined in a rectangular region of $400 \times 400$ km² corresponding to the area shown in Figure 1. By use of the mentioned C-Sharp application, the barrier zones were marked on the area (the blue color in Figure 1).

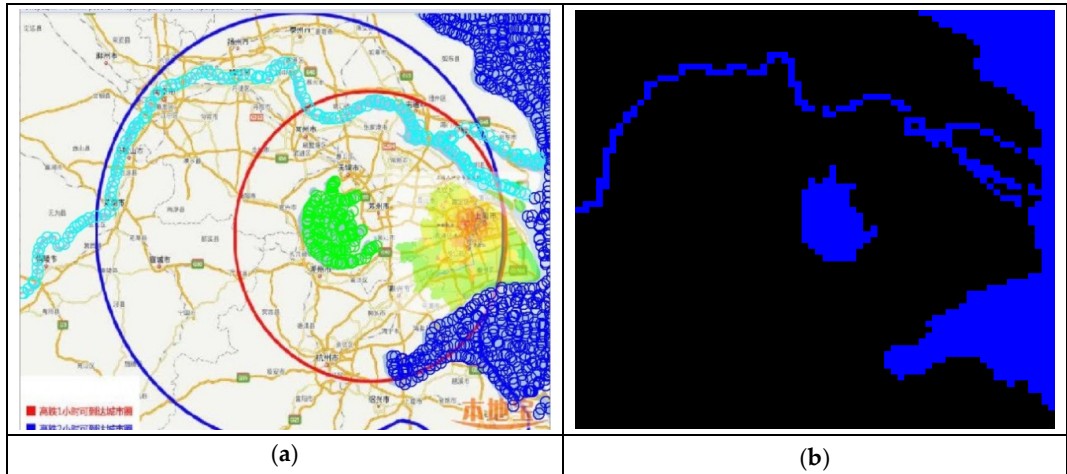

|     |     |
| --- | --- |
| (a) | (b) |

**Figure 1.** Processing the 2017 data for Shanghai and surrounding areas ($400 \times 400$ km), taking into account the building density and landscape features [27]: (**a**) Inside the red circle is the area of high building density, and the belt from the red to the blue circle is the area of lower building density; (**b**) distribution of water (blue color) and land (black color) in the territory, given by the raster data model corresponding to (a).

The initial distribution of the activator, $u$, was taken according to data on the population density distribution in 2017, shown in Figure 2.

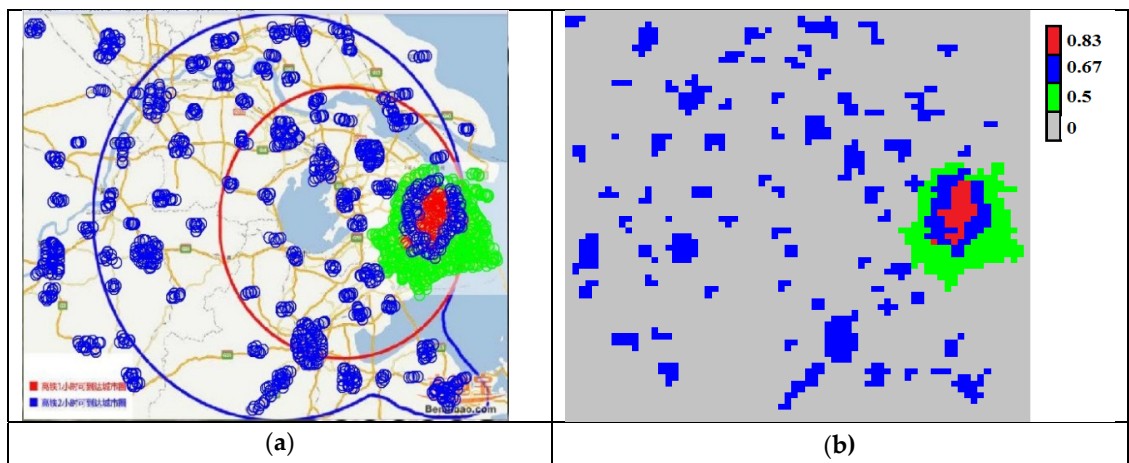

| (a) | (b) |

**Figure 2.** Population density zones of Shanghai and surrounding areas in 2017 ($400 \times 400$ km): (**a**) Red marks the Shanghai core with the maximum population density (blue and green color denote reduced population density in Shanghai); the zone inside the red circle is the near suburbs, while the belt from the red circle to the blue circle is distant suburbs; (**b**) the distribution of the building density as a fraction per square kilometer, given by the raster data model corresponding to (a).

We collected data on the housing cost [51] in the central areas of Shanghai and its recently joined districts [52] and in adjacent territories from 2015 to 2017; we then used these data to calculate housing cost growth rates in the considered areas. Firstly, the annual increase in housing cost as a percentage was calculated, and this was then averaged over the years. The results are shown in Chart 4 below.

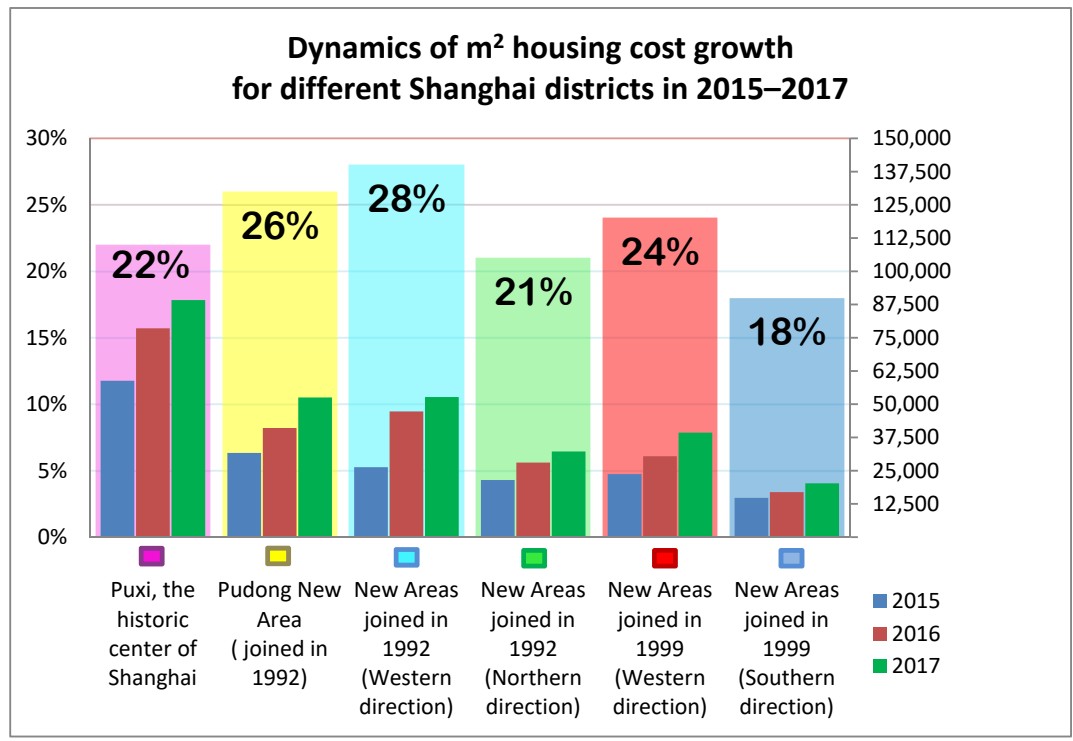

**Chart 4.** Dynamics of m$^2$ housing cost growth for different Shanghai districts in 2015–2017 [27,51].

We used these data to obtain the raster data model of the distribution of the parameter $\gamma(x,y)$ by means of the abovementioned C Sharp application. The results are shown in Figure 3. The colors of the rectangles matching data in percentages in Chart 4 correspond to the colors in Figure 3.

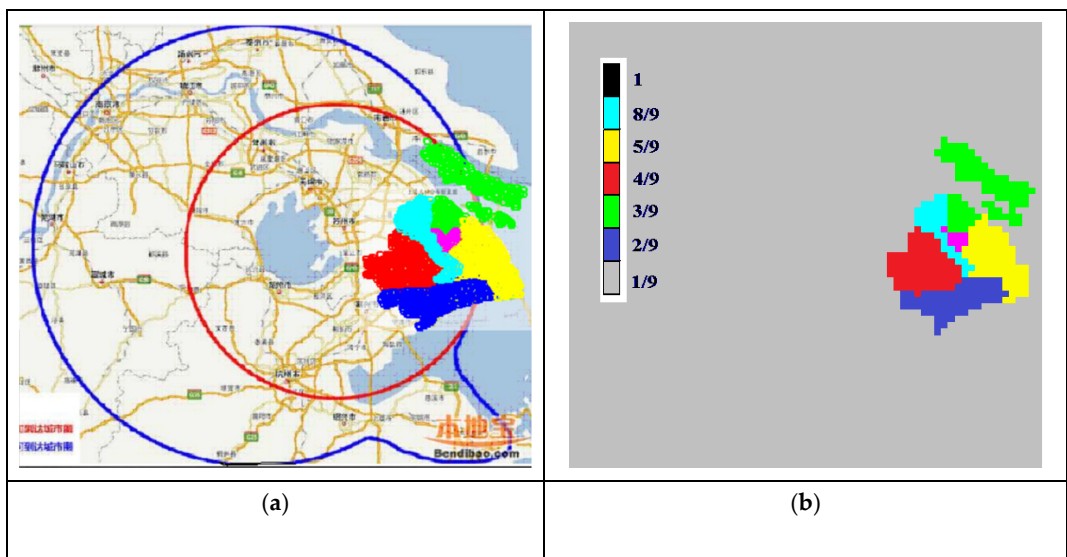

**Figure 3.** Dynamics of cost growth (housing m²/year) for different Shanghai districts in 2015–2017 [27,51]: (**a**) data from processing Chart 4; (**b**) the housing cost distribution as a share of the maximum (in %) in the raster data model corresponding to (**a**).

As parameter $\gamma$ corresponds to the dependence of the m² housing cost on the building area in 2015–2017, the initial distribution of the inhibitor $v$ is the same as the distribution of $\gamma$ and can be taken from Chart 4 (Figure 3).

### 2.3.2. The Choice of Model Parameters

Based on the data for 2005–2010 from [31] and Charts 2 and 3, we assumed $D_u$ = 5066 ha/year inside the red circle (Figure 1) and $D_u$ = 929 ha/year inside the blue circle (Figure 1).

From data analysis in Chart 4, we concluded that in 2017, relative to 2016, the housing cost rose by 22% [27,51]. That is, in 2017, the same budget was enough to build only 78% of the realty from the 2016 amount, so in the model we set $D_v = 0.22\, D_u$.

The function $\alpha(x,y)$ in Equation (1) is the activation parameter. An analysis of the system of Equation (1) [43–46] showed that with $\alpha$ = 0.3, the system describes an excitable medium, and with $\alpha$ = 1, it describes a non-excitable medium. In the model, we put $\alpha$ = 0.3 in the East direction. This numerical value was determined from regulatory acts of China, according to which 30% of geobiocoenoses should be maintained in developed areas. We put $\alpha$ = 1 in the non-excitable zones of the sea, Lake Taihu, and the Yangtze River.

### 2.3.3. The Method of Numerical Solution

The solution was determined by means of evolutionary factorization in the domain corresponding to Figure 1. On the boundaries of the domain, homogeneous Neumann conditions were set. The iterative process was continued until complete stationing. Distributed computing was performed using AMD graphics processors and the OpenCL compiler.

## 3. Results

Analysis performed by means of the developed model (Figure 4) showed that preservation of zoning existed by 2017 (Figure 2) and, therefore, a differential approach to housing cost will allow us to not only meet the demand for housing of a large amount of people, but to also develop steadily in accordance with the standards of natural geobiocoenosis conservation. This will reduce the population density in the main Shanghai territory (red circle in Figure 1) and will promote economic development in the territories of near and far suburbs. This forecast corresponds to the development plans of Shanghai.

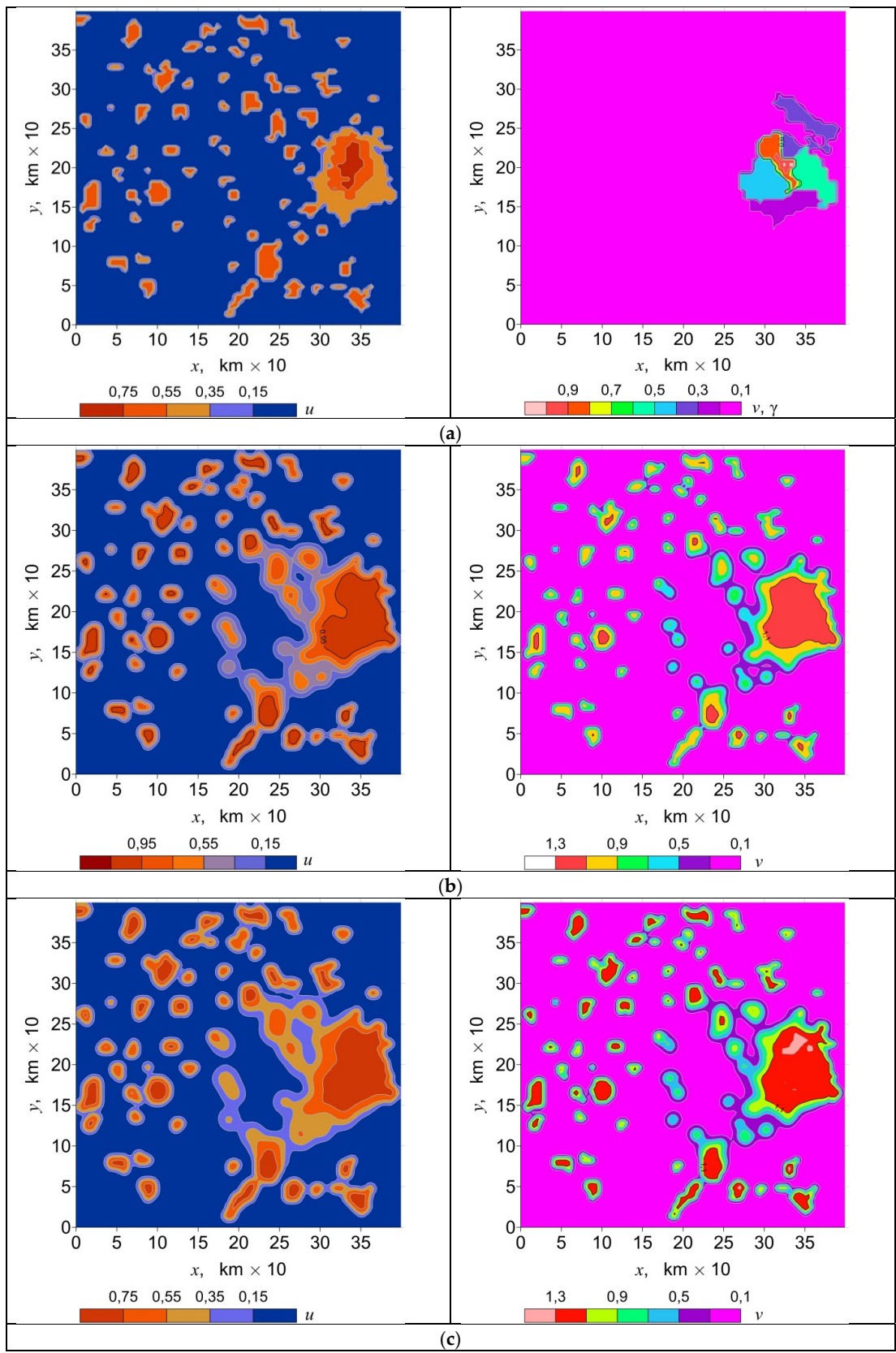

**Figure 4.** Model of Shanghai development on the basis of building dynamics and housing cost: (**a**) 2017—the initial conditions; (**b**) 2020, (**c**) 2030. $u$ is the building area (share of km$^2$, dimensionless value); $v$ is the 1 m$^2$ housing cost (share of the maximum possible housing cost in 2017, dimensionless value).

## 4. Discussion

The model is based on the general principles of active medium dynamics; therefore, it is universal for any object that can be considered an active medium. The only difference when using the model to predict the development of urban ecosystems in countries with different levels of socio-economic development and different political prerequisites is the variety of parameters included in the model: The activation parameter (function $\alpha(x,y)$ in Equation (1)), determining the autowave process inhibitors, and the characteristic scales of the activator and inhibitor. Also, cartographic and statistical data are essential. The model is based on national urban development programs, so private financial and economic risks in no way affect the overall picture, and in conditions of sustainable socio-economic development, the model is predictive.

### 4.1. Model Reliability

Earlier, we used this model to describe the expansion of some Moscow areas. In this case, we chose the standard values of green planting areas, which should be preserved in the megacity environment as an inhibitor [53]. The model was tested on the example of the Fili area of Moscow's expansion in 1952–1968, and the results showed good agreement with real data. The results of the comparison are presented in Figure 5.

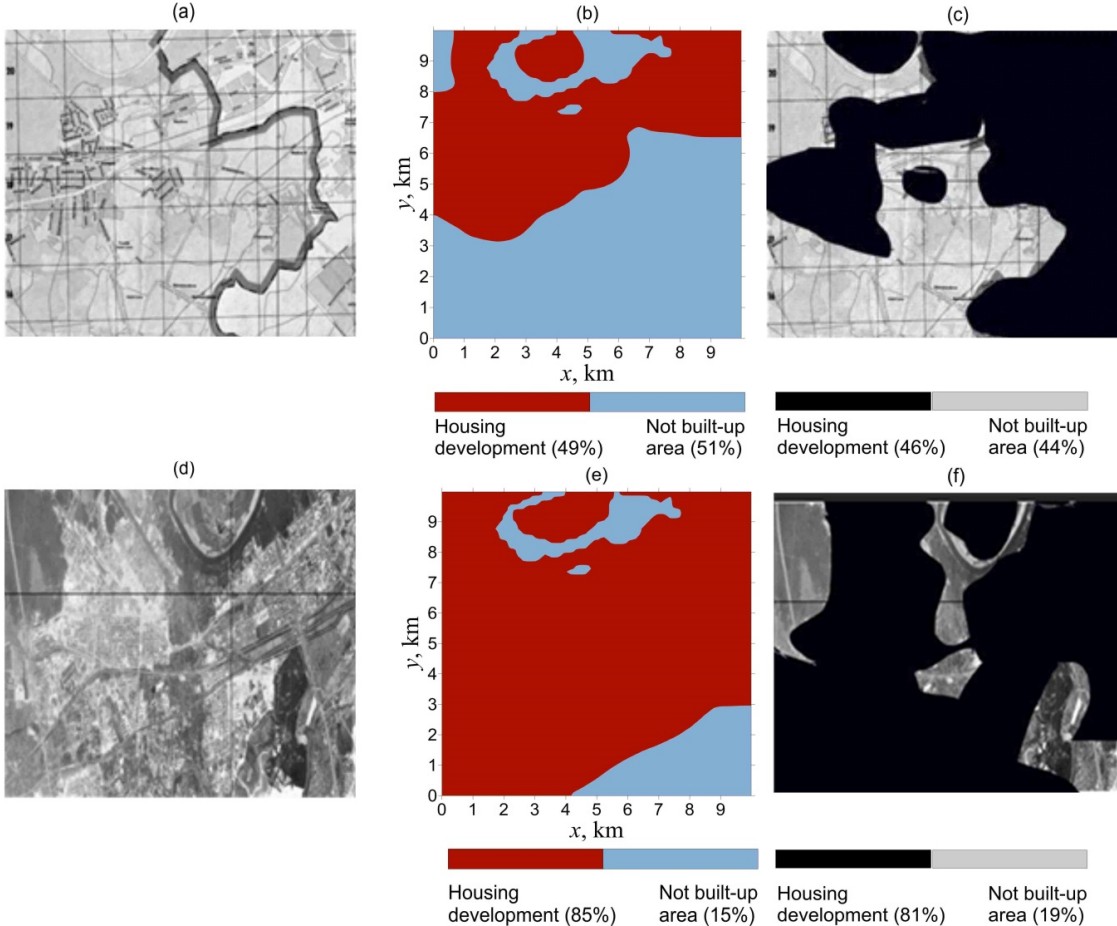

**Figure 5.** Expansion of Moscow near the Moscow River floodplain in 1952–1968: (**a**) 1960; a map of the considered region [54], (**b**) 1960; model calculations, (**c**) 1960; a map of the considered region with marked urban areas (building density over 55%); (**d**) 1968; a map of the considered region [54], (**e**) 1968; model calculations, (**f**) 1968; a map of the considered region with marked urban areas (building density over 55%).

To evaluate the area of urban development on the basis of map data [54], Adobe PS was used to mark the areas with built-up area greater than 55% black on the maps. Then, by means of Adobe PS, the percentage ratio of black marked areas to the total area was calculated.

A comparison of the obtained data with aerial photographic data of the studied areas in 1960 and 1968 showed that the modeling error was less than 10%.

### 4.2. Development Outlook for Shanghai and the Surrounding Areas

According to our calculations (Figure 4), the area of Shanghai and the surrounding territory with a building density of more than 55% (urban area) in the domain under consideration currently occupies 11% of the territory. That is 17,600 km$^2$. Based on the model, we conclude that by 2030, the area of dense urban development will increase by 27% and will be 24,000 km$^2$, which is 15% of the territory. At the same time, the housing cost will increase by 34%.

## 5. Conclusions

According to our concept, an urban ecosystem can be considered an active medium with interacting natural and anthropogenic subsystems characterized by considerable heterogeneity of the anthropogenic influence sources and urban geobiocoenosis. Based on this view, an autowave model of megacity development was offered. Using the example of Moscow expansion in the period from 1952 to 1968, a comparison was made of the model results with real data. The difference was less than 10%.

The model can be used to predict the development of megacities in various countries around the world. When used for this purpose, preliminary analysis should be conducted to determine the activators and inhibitors of megalopolis development and the characteristic scales.

By means of the model, a prediction of the development of Shanghai and the surrounding territory until 2030 was made. According to our forecast, the urban area of the territory will increase in the near future, as will the housing cost.

**Author Contributions:** Conceptualization A.S. (Alla Sidorova), N.L., and M.N.; methodology, N.L. and A.S. (Alla Sidorova); software, N.L. and A.S. (Anna Semina); validation, A.S. (Anna Semina) and M.N.; formal analysis, A.S. (Alla Sidorova); investigation, A.S. (Anna Semina) and M.N.; writing—original draft preparation, A.S. (Alla Sidorova); writing—review and editing, N.L.; visualization, N.L. and A.S. (Alla Sidorova); supervision, A.S. (Alla Sidorova); project administration, A.S. (Alla Sidorova); funding acquisition, N.L.

**Funding:** The research was carried out within the framework of the Russian Science Foundation, project No. 18-11-00042.

**Acknowledgments:** The authors are sincerely grateful to the reputable anonymous reviewers for the most valuable comments that allowed us to make the article logically complete. The authors are also grateful to MDPI English Editing for the he fast and qualified text verification.

**Conflicts of Interest:** The authors declare no conflict of interest.

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
