# Peer review of "A Spatio-Temporal Autowave Model of Shanghai Territory Development"

_sustainability, doi:10.3390/su11133658_

Round 1
Reviewer 1 Report
The article addresses an important issue in a
setting that deserves more attention. Despite my great interest in the subject,
I find the manuscript has three most frustrating aspects. First, its deficiency
in theoretical development which doesn’t properly situate the study in
the context of the larger literature on urban development modeling/prediction;
second, the vague methodological section which makes the conclusions less than convincing;
and third, its poor narrative structure and writing quality which detract
significantly from the paper's clarity and impact.
To sum up, there is not enough theoretical development or methodological validation in this manuscript to warrant publication as a research article.
Author Response
The article has been substantially revised. The bibliography has been more than doubled. A detailed he authors thank the distinguished reviewer for the attention to our work and the valuable comments made.
literature review was made. In the new version of the article, the introduction contains a detailed description of the physical prerequisites for creating the proposed model, as well as links to articles that substantiate our approach from a mathematical point of view. In addition, a paragraph was included with the results of testing our model.
Reviewer 2 Report
The aims are described in the Introduction to the text: "to build an autowave model for the development of Shanghai, taking into account the basic mechanisms of this process: the spatial and socio-economic feasibility of development, the dynamics of housing prices, the proportion of land under agriculture and the geographical features of adjacent territories." This paper is considered interesting although it presents certain gaps that generate doubts about the results obtained, especially regarding the variables used in the study, and especially those that have not been taken into consideration.
The growth prediction model of the city of Shanghai used, seems to be an excessive simplification of urban reality, not integrating at all much of the spatial complexity of the surrounding territory. The paper does not recognize such limitations at all.
The following comments and suggestions are specifically made:
(1) The abstract must specify how the "Fitz-Hugh-Nagumo system of equations" is modified.
(2) The paper uses variables such as "the socio-economic feasibility of development, the dynamics of housing prices, the share of land for agriculture and landscape features". All the specific variables used should be clearly specified, perhaps by means of a table with the bibliographical references that support it. Also explain why these variables are used and not others. In any case, an excessively simplistic view of urban processes is observed, apparently without sufficient bibliographic support.
(3) Urbanization processes should not be understood exclusively as an "autocatalytic" process, it is much more complex. there are certain externalities to the urbanization process itself that are key. For example: risk management of various kinds, geological suitability, administrative management of different areas, presence or absence of mobility infrastructure, proximity to other cities on a regional scale, geopolitical or administrative boundaries, etc. None of these variables seem not to be considered in the paper, nor recognized as a limitation of the research or the model.
(4) It is recommended that you explain the advantages, and disadvantages, that this model provides compared to others, such as those based on ANN as are the Self Organizing Maps. The discussion does not describe any limitation or disadvantage of the model.
(5) The data contained in Tables 1, 2, 3 and 4 could be understood more easily and immediately with graphs, some of which could be overlayed.
Explain the meaning of RS. No such reference appears in the quoted text.
(7) The meaning of the use of an "authoring application that allows you to create text files based on images" is not well understood. Does it refer to the rasterization of information? What the algorithm does must be clearly explained. If the algorithm measures density, it must explain exactly what type of density (construction, urbanization, population, etc.) In any case that information is probably already available in the GIS systems of the region, it is not clear the need for such implementation in the methodology.
(8) There are fewer key references on the concept of "Ecosystem services" that denote the complexity of the concept, for example: Bolund, P., & Hunhammar, S. (1999). Ecosystem services in urban areas. Ecological economics, 29(2), 293-301.
(9) Table 4 and Figure 3 show 6 types of "Dynamics of cost growth". How they have been obtained should be shown methodologically. The meanings of these clusters are unknown.
(10) Lines 144-145: "Shanghai is a flood plain without obvious changes in altitude and gradient, so the landscape features that inhibit the development of many cities are not taken into account in this case".
To estimate that because the territory is "flood plain" there are no landscape elements worthy of being taken into account in the development of a territory is reckless to say the least. For example, it stops considering any notion of landscape heritage, environmental value, etc. This absence obviously implies a limitation that must be considered or at least expressly declared in the article, both in the body of the paper and in the abstract.
(11) The identity variables of the territory under study that influence the model should be made explicit. It seems obvious that in the growth model, the spatial qualities and identities of the same should be considered. Otherwise, the model does not cease to be an abstract model that does not take into consideration the place where it is implemented, losing a clear value for the planning and real prediction of future events.
(12) In the discussion, apart from the reference to Moscow, other references indicating the differential value of the proposed methodology should be incorporated.
Author Response
The authors are sincerely grateful to the distinguished reviewer for the most valuable comments that allowed us to make the article logically complete
(1) The abstract must specify how the "Fitz-Hugh-Nagumo system of equations" is modified.
Due to the distinguished reviewer comments the material delivery concept has been significantly changed as well as the abstract. The Fitz-Hugh-Nagumo system was excluded from abstract/ the mentioned explanation is given in introduction, namely the text:
“To create a model describing such a complex and nonlinear object as UES we made modifications to this system, changing it so that it could describe not only propagating fronts, but also stationary solutions with large gradients at the boundaries of barriers (natural geobiocenoses that prevent autowave motion). The conditions for existence of such solutions are formulated in [22,23]. In addition, a cross-product component was added to the equation for the activator enhancing the feedback between the activator and the inhibitor. The theoretical basis of our investigation is the analysis of the UES stability on the basis of synergetic ideas on the autowave self-organization in conjugated active media [24-27].”
(2) The paper uses variables such as "the socio-economic feasibility of development, the dynamics of housing prices, the share of land for agriculture and landscape features". All the specific variables used should be clearly specified, perhaps by means of a table with the bibliographical references that support it. Also explain why these variables are used and not others. In any case, an excessively simplistic view of urban processes is observed, apparently without sufficient bibliographic support.
Unnecessary details containing specific terminology and leading away from the essence of study have been eliminated
(3) Urbanization processes should not be understood exclusively as an "autocatalytic" process, it is much more complex. there are certain externalities to the urbanization process itself that are key. For example: risk management of various kinds, geological suitability, administrative management of different areas, presence or absence of mobility infrastructure, proximity to other cities on a regional scale, geopolitical or administrative boundaries, etc. None of these variables seem not to be considered in the paper, nor recognized as a limitation of the research or the model.
The model contains parameters that indirectly take into account geographical factors, since such development is possible only in these favorable conditions. Our model is based on the national urban development program of China, so private financial economic risks do not affect the overall picture. If the socio-demographic dynamics of China’s development fit the government’s plans, our model will provide an objective picture, any disasters that are turbulences of economic or natural factors should be described by models of another plan. Corresponding explanations were included in the text of the article
(4) It is recommended that you explain the advantages, and disadvantages, that this model provides compared to others, such as those based on ANN as are the Self Organizing Maps. The discussion does not describe any limitation or disadvantage of the model.
The following paragraph was included in the text:
“The model is based on the general principles of the active media dynamics; therefore, it is universal for any objects that can be considered as an active medium. The only difference when using the model to predict the development of urban ecosystems in countries with different levels of socio-economic development and different political prerequisites is variety of parameters included in the model that are the activation parameter (function α(x,y,t) in system (1)), determining of the autowave processes inhibitors and the characteristic scales of activator and inhibitor. Also cartographic and statistical data are essential. The model is based on national urban development programs, so private financial and economic risks in no way affect the overall picture and in conditions of sustainable socio-economic development the model is predictive.”
(5) The data contained in Tables 1, 2, 3 and 4 could be understood more easily and immediately with graphs, some of which could be overlayed.
The authors are especially grateful for this advice. We have made the diagrams on base of the tables. the diagrams are much more visual.
(6) Explain the meaning of RS. No such reference appears in the quoted text.
That seems to be “remote sensing”, the term was excluded as there wasn’t any explanation
(7) The meaning of the use of an "authoring application that allows you to create text files based on images" is not well understood. Does it refer to the rasterization of information? What the algorithm does must be clearly explained. If the algorithm measures density, it must explain exactly what type of density (construction, urbanization, population, etc.) In any case that information is probably already available in the GIS systems of the region, it is not clear the need for such implementation in the methodology.
The following paragraph was included into “Materials and methods”:
2.2. Map Data Processing
To analyze the cartographic data, the authors (An.S.) created a C-Sharp code which makes it possible to translate information from graphic images into digital data. For this purpose, points are plotted on the map that differ in color depending on the properties of a part of the territory (for our needs, this is either the density of the building, or the rate of growth in housing prices, or the presence of water barriers). Then each color is associated with certain numerical value of the parameter included in the model. These values are stored in binary arrays, and then used in the program code for describing the UES dynamics
(8) There are fewer key references on the concept of "Ecosystem services" that denote the complexity of the concept, for example: Bolund, P., & Hunhammar, S. (1999). Ecosystem services in urban areas. Ecological economics, 29(2), 293-301.
The article was cited
(9) Table 4 and Figure 3 show 6 types of "Dynamics of cost growth". How they have been obtained should be shown methodologically. The meanings of these clusters are unknown.
The data was taken from the realty site, ref [45]: Distribution of prices per square meter of housing in Shanghai. Available online: http://m.focus.cn/sh/daogou/11042556 (accessed on 15 July 2018). The methodic of data obtaining was explained
(10) Lines 144-145: "Shanghai is a flood plain without obvious changes in altitude and gradient, so the landscape features that inhibit the development of many cities are not taken into account in this case". To estimate that because the territory is "flood plain" there are no landscape elements worthy of being taken into account in the development of a territory is reckless to say the least. For example, it stops considering any notion of landscape heritage, environmental value, etc. This absence obviously implies a limitation that must be considered or at least expressly declared in the article, both in the body of the paper and in the abstract.
The following paragraph was included in the text: “Shanghai is a flood plain without obvious changes in altitude and gradient. Studying of atmospheric factors like precipitation and temperature in Shanghai and surrounding areas [47,48] showed that in general they did not have an impact on Shanghai borders expansion. Therefore, the most significant factors defining the territory expansion are the number and density of potential buyers of housing and its cost. We must also take into account the landscape features mainly related to the presence of water bodies.”
(11) The identity variables of the territory under study that influence the model should be made explicit. It seems obvious that in the growth model, the spatial qualities and identities of the same should be considered. Otherwise, the model does not cease to be an abstract model that does not take into consideration the place where it is implemented, losing a clear value for the planning and real prediction of future events.
Our model is based on specific values of development areas and real estate prices and takes into account the spatial distribution of these parameters, as well as the normative ones specified in the regulatory acts. In addition, in the new version of the article we included more specific calculations (in km2) regarding the predicted building areas.
(12) In the discussion, apart from the reference to Moscow, other references indicating the differential value of the proposed methodology should be incorporated.
The following paragraph was included in “Conclusion”:
“The model can be used to predict forecasts for the development of megacities in various countries of the world. For this purpose the preliminary analysis is to be conducted to determine the activators and inhibitors of megalopolis development processes and the characteristic scales.” See also the remark (4) of current revision
Reviewer 3 Report
The methodology framework of the paper is excellent and innovative. However, for the target issue of Sustainability, this paper fails to discuss the practical contributions as well as the connections and contributions to the previous works. The authors need to re-frame the introduction, discussions and conclusion sections to reflect these changes. Moreover, I suggest the addition of a literature review session because the paper is currently too short for a regular research paper in this field. Usually a paper is expected to have 400-500 lines and 40-50 references. This paper is very short because it does not discuss previous literature and practical contributions that are related to the methodology.
Detailed comments:
Line 25: This is written in an unprofessional way. In the abstract, “the processing of cartographic data” should be provided by a terminology, not a description written like user-manual.
Line 34: If you present an excellent methodology work, why you start with a discussion on population and Shanghai? That is a total downgrade of your methodology work into a case study. The authors should discuss how their works relate to previous methods of land use and urban growth model, such as:
Do you improve the accuracy and usability of land use models?
https://link.springer.com/article/10.1007/s12061-019-09296-5
How is your model connected and compared to the emerging machine learning application in urban growth models?
https://www.sciencedirect.com/science/article/pii/S0198971516302265
Line 82: Please be confident that your method can be applied worldwide and has many more potentials – make these discussions and draw international case studies. Shanghai case study is not interesting to most readers if they do not focus on China issues. I have read way too much Shanghai and way too less on the methodology discussion till this point in the introduction –please relegate Shanghai discussions to no more than 1 short paragraph in the introduction.
Table 4: Again, for readers not familiar with Shanghai, a long table with district names does not have much readability. A spatial distribution of housing price showing its concentration in urban centers/subcenters/suburban areas is much more valuable.
Line 176: Although I like your method, there is still a need to show to the general audience the applicability of Fitzhugh-Nagumo system for your modeling goal. In what practical situation the model is applied? Why do you think it is fitted to urban growth modeling based on previous applications?
Line 211: Result presentation is too short and un-inspiring. The first the authors need to discuss the general spatial distribution of their forecasts – are the growth occur more in-fill, on urban fringe, or leapfrogging? Is it compact or not? Also, comparison to official plans should be more explicit – in what ways they are similar or dissimilar? Do you have explanations or intuitions for these?
Discussions and conclusions: Again, these parts fail to show the value of your model to the target readers of this journal. This part should extend your discussions a bit to show your model values to the practicality and connections to previous literature, for example:
Urban growth and climate impacts
https://journals.ametsoc.org/doi/full/10.1175/JCLI4109.1
Urban growth model and ecosystem service impact:
https://www.sciencedirect.com/science/article/pii/S1470160X19301682
Author Response
The authors are sincerely grateful to the distinguished reviewer for the most valuable comments that allowed us to make the article logically complete
The methodology framework of the paper is excellent and innovative. However, for the target issue of Sustainability, this paper fails to discuss the practical contributions as well as the connections and contributions to the previous works. The authors need to re-frame the introduction, discussions and conclusion sections to reflect these changes. Moreover, I suggest the addition of a literature review session because the paper is currently too short for a regular research paper in this field. Usually a paper is expected to have 400-500 lines and 40-50 references. This paper is very short because it does not discuss previous literature and practical contributions that are related to the methodology.
References have been significantly expanded. The references mentioned by a distinguished reviewer were included
Detailed comments:
Line 25: This is written in an unprofessional way. In the abstract, “the processing of cartographic data” should be provided by a terminology, not a description written like user-manual.
The text was excluded from “Abstract”. The following paragraph was included in “Materials and methods”: 2.2. Map Data Processing
“To analyze the cartographic data, the authors (An.S.) created a C-Sharp code which makes it possible to translate information from graphic images into digital data. For this purpose, points are plotted on the map that differ in color depending on the properties of a part of the territory (for our needs, this is either the density of the building, or the rate of growth in housing prices, or the presence of water barriers). Then each color is associated with certain numerical value of the parameter included in the model. These values are stored in binary arrays, and then used in the program code for describing the UES dynamics”
Line 34: If you present an excellent methodology work, why you start with a discussion on population and Shanghai? That is a total downgrade of your methodology work into a case study. The authors should discuss how their works relate to previous methods of land use and urban growth model, such as:
Do you improve the accuracy and usability of land use models?
https://link.springer.com/article/10.1007/s12061-019-09296-5
How is your model connected and compared to the emerging machine learning application in urban growth models?
https://www.sciencedirect.com/science/article/pii/S0198971516302265
The model proposed is based on the theory of active media. After these valuable comments of the distinguished reviewer we have changed the concept of the paper and elaborated on the properties and prerequisites of the model in the introduction.
In our case, a process similar to “learning” in the model of cellular automata is a preliminary analysis of the object of study and the inclusion in the model of specific parameters that characterise selected metropolis. Related to this the following paragraph was included:
“In order to effectively apply the activator-inhibitor model for an adequate description of urban ecosystems, it is necessary to thoroughly understand the reasons for their growth and the processes that inhibit this growth. While the density of population and buildings is an obvious activator of urban ecosystem development, the inhibitor is determined by the policy determining the urban planning and is different for each country [29-30]. Thus, any study should be preceded by the choice of the object and the study of its individual characteristics”
Line 82: Please be confident that your method can be applied worldwide and has many more potentials – make these discussions and draw international case studies. Shanghai case study is not interesting to most readers if they do not focus on China issues. I have read way too much Shanghai and way too less on the methodology discussion till this point in the introduction –please relegate Shanghai discussions to no more than 1 short paragraph in the introduction.
The following paragraph was included:
“The model is based on the general principles of the active media dynamics; therefore, it is universal for any objects that can be considered as an active medium. The only difference when using the model to predict the development of urban ecosystems in countries with different levels of socio-economic development and different political prerequisites is variety of parameters included in the model that are the activation parameter (function α(x,y,t) in system (1)), determining of the autowave processes inhibitors and the characteristic scales of activator and inhibitor. Also cartographic and statistical data are essential. The model is based on national urban development programs, so private financial and economic risks in no way affect the overall picture and in conditions of sustainable socio-economic development the model is predictive.”
Table 4: Again, for readers not familiar with Shanghai, a long table with district names does not have much readability. A spatial distribution of housing price showing its concentration in urban centers/subcenters/suburban areas is much more valuable.
We have changed the tables by diagrams incidentally removing unnecessary specification
Line 176: Although I like your method, there is still a need to show to the general audience the applicability of Fitzhugh-Nagumo system for your modeling goal. In what practical situation the model is applied? Why do you think it is fitted to urban growth modeling based on previous applications?
The authors are especially grateful to the distinguished reviewer for this remark. The new version of “Introduction” in devoted to explain this.
Line 211: Result presentation is too short and un-inspiring. The first the authors need to discuss the general spatial distribution of their forecasts – are the growth occur more in-fill, on urban fringe, or leapfrogging? Is it compact or not? Also, comparison to official plans should be more explicit – in what ways they are similar or dissimilar? Do you have explanations or intuitions for these?
We have expanded the results explanation and included the the reliability of the model proving.
Discussions and conclusions: Again, these parts fail to show the value of your model to the target readers of this journal. This part should extend your discussions a bit to show your model values to the practicality and connections to previous literature, for example:
Urban growth and climate impacts
https://journals.ametsoc.org/doi/full/10.1175/JCLI4109.1
Urban growth model and ecosystem service impact:
https://www.sciencedirect.com/science/article/pii/S1470160X19301682
The model contains parameters that indirectly take into account geographical factors, since such development is possible only in these favorable conditions.
The following paragraph was included:
“The model is based on national urban development programs, so private financial and economic risks in no way affect the overall picture and in conditions of sustainable socio-economic development the model is predictive.”
Also we have carefully studied all mentioned papers, essentially expanded the literature review and described in detail the background and theoretical justification of our model in the introduction.
Round 2
Reviewer 1 Report
The article addresses an important issue in a setting that deserves more attention. Despite my great interest in the subject, I find the manuscript has three most frustrating aspects. First, its deficiency in theoretical development which doesn’t properly situate the study in the context of the larger literature on urban development modelling/prediction; second, the vague methodological section which makes the conclusions less than convincing; and third, its poor narrative structure and writing quality which detract significantly from the paper's clarity and impact.
To sum up, there is not enough theoretical development or methodological validation in this manuscript to warrant publication as a research article.
Author Response
The authors are thankful to distinguished Reviewer for valuable comments.
I find the manuscript has three most frustrating aspects.
First, its deficiency in theoretical development which doesn’t properly situate the study in the context of the larger literature on urban development modelling/prediction;
The paragraph “2.2. Theoretical background of spatio-temporal model of Shanghai development” was created explaining the theoretical prerequisits of the model
second, the vague methodological section which makes the conclusions less than convincing;
A paragraph “2.3. The Metodology of Study” was created, where we described in detail the methodology of the study.
and third, its poor narrative structure and writing quality which detract significantly from the paper's clarity and impact.
As far as possible, the text was freed from complex terms and long sentences.
Reviewer 2 Report
In general terms the article has improved notably with the changes made, gaining in clarity.
The following comments and suggestions are specifically made:
(1) It is recommended to review the "Chart 3" since the complete legend is not shown, the meaning of the red and green bars is not known.
(2) In section "Map Data Processing" is described the methodology for creating a map in which the colour of each point represents specific properties of the territory. This type of map in GIS is called "raster data model". It is recommended to incorporate such a concept that is widely known in urban and territorial planning and therefore common in journals such as Sustainability.
(3) Line 162. It is not considered necessary to indicate here the authorship of the software code. It is already indicated under "Author Contributions".
(4) Lines 210-211. Review text "We must also take into account the landscape features mainly related to the presence of water bodies".
Author Response
The authors are thankful to distinguished Reviewer for valuable comments.
(1) It is recommended to review the "Chart 3" since the complete legend is not shown, the meaning of the red and green bars is not known.
That was improved
(2) In section "Map Data Processing" is described the methodology for creating a map in which the colour of each point represents specific properties of the territory. This type of map in GIS is called "raster data model". It is recommended to incorporate such a concept that is widely known in urban and territorial planning and therefore common in journals such as Sustainability.
The phrase "map data processing" was changed to "raster data model".
(3) Line 162. It is not considered necessary to indicate here the authorship of the software code. It is already indicated under "Author Contributions".
That was improved
(4) Lines 210-211. Review text "We must also take into account the landscape features mainly related to the presence of water bodies".
This sntence was moved to the paragraph, which explains what in the model should be considered as barriers
Reviewer 3 Report
The new version has addressed reviewer concerns.
Author Response
The authors are thankful to distinguished Reviewer
Round 3
Reviewer 1 Report
The authors have addressed most of my previous concerns. I would recommend citing the following research in the introduction section of this paper: Li, C., Song, Y., & Chen, Y. (2017). Infrastructure development and urbanization in China. In China’s urbanization and socioeconomic impact (pp. 91-107). Springer, Singapore. Also, it is recommended that the authors perform a thorough edit to ensure proper grammar and accurate citation. The authors may need to find a native English speaker to edit the manuscript.
Author Response
The authors thank the reputable Reviewer for valuable recommendations
I would recommend citing the following research in the introduction section of this paper: Li, C., Song, Y., & Chen, Y. (2017). Infrastructure development and urbanization in China. In China’s urbanization and socioeconomic impact (pp. 91-107). Springer, Singapore.
The article was cited
Also, it is recommended that the authors perform a thorough edit to ensure proper grammar
The English edition was performed by MDPI English Editing
and accurate citation.
All cites were checked, the bugs were fixed